# Wide-Angle Beam Steering Closed-Form Pillbox Antenna Fed by Substrate-Integrated Waveguide Horn for On-the-Move Satellite Communications

**DOI:** 10.3390/s24030732

**Published:** 2024-01-23

**Authors:** Muhammad Ikram, Kamel Sultan, Ahmed Toaha Mobashsher, Mahdi Moosazadeh, Amin Abbosh

**Affiliations:** 1School of Electrical Engineering and Computer Science, The University of Queensland (UQ), Brisbane 4072, Australia; m.ikram@uq.net.au (M.I.); a.mobashsher@uq.edu.au (A.T.M.); m.moosazadeh@uq.edu.au (M.M.); a.abbosh@uq.edu.au (A.A.); 2College of Engineering and Applied Science (CEAS), American University of Kuwait, Safat 13034, Kuwait

**Keywords:** flat panel antenna, low earth orbit satellites, on-the-move communications

## Abstract

Wide-angle mechanical beam steering for on-the-move satellite communications is presented in this paper based on a closed-form pillbox antenna system. It includes three main parts: a fixed-feed part, which is a substrate-integrated waveguide (SIW) horn with an extended aperture attached to a parabolic reflector; a novel quasi-optical system, which is a single coupling slot alongside and without spacing from the parabolic reflector; and a radiating disc, which is a leaky-wave metallic pattern. To make the antenna compact, pillbox-based feeding is implemented underneath the metallic patterns. The antenna is designed based on a substrate-guided grounded concept using leaky-wave metallic patterns operating at 20 GHz. Beam scanning is achieved using mechanical rotation of the leaky-wave metallic patterns. The proposed antenna has an overall size of 340 × 335 × 2 mm^3^, a gain of 23.2 dBi, wide beam scanning range of 120°, from −60° to +60° in the azimuthal plane, and a low side lobe level of −17.8 dB at a maximum scan angle of 60°. The proposed antenna terminal is suitable for next-generation ubiquitous connectivity for households and small businesses in remote areas, ships, unmanned aerial vehicles, and disaster management.

## 1. Introduction

Recently, there has been a rise in the demand for satellite services primarily driven by the necessity for worldwide broadcasting and the establishment of networks for sharing data information [1,2,3]. These networks include digital radio, television, and broadband internet services. To meet the requirements of those services, the integration of wireless communications technology with low earth orbit (LEO) satellite communications systems is introduced [3,4,5,6]. So, a low-cost and compact antenna is required to connect with LEO satellites. This antenna can be mounted on ambulances, unmanned aerial vehicles (UAVs), trains, vehicles, ships, robots for remote sensing, and homes to provide satellite communications on-the-move (SCOM) services (see Figure 1). While the traditional parabolic antenna is a high-performance solution, its large size makes it unsuitable for most of those applications. Therefore, there is a growing need for flat, compact, lightweight, and mechanically robust antennas that align with the specific requirements of these systems [7,8,9,10,11]. Additionally, the antennas’ gain and beam coverage range play a significant role in overcoming path loss in high-frequency bands and enhancing resolution [12,13,14,15,16,17,18,19]. For SCOM systems, it is essential to have an antenna that can balance high gain and a wide coverage area. Additionally, the antenna must be capable of beam steering to establish real-time satellite-to-ground connections for SCOM [20]. This is where the beam steering capability of the antenna plays an essential role.

Beam-scanning antenna arrays can use either mechanical or electrical scanning approaches [21,22]. Electrical beam scanning can be achieved through switched antenna arrays or phased antenna arrays. Although these methods offer the advantages of low profile and rapid scanning speed, they are expensive, complex to implement, and suffer from high loss at higher frequencies, making them unsuitable for use in commercial terminal devices [23,24,25,26]. On the other hand, mechanical beam scanning is low-cost and offers high gain, but it is typically bulky and heavy and unsuitable for recent applications requiring compact designs [21,27].

Several different designs have been presented for beamforming feeding networks. These designs are based on passive technologies and do not rely on active components. Examples of these designs include the Rotman lens [28,29], Luneburg lens [12,30,31], Butler matrix [32,33], and pillbox reflector [34,35,36]. Pillbox reflector-based feeding designs can steer the beam across a wide bandwidth, but they require multiple feeding ports and provide limited beam coverage. The 3D-printed lens is unsuitable for applications requiring low-profile and conformal features as it has a high profile and fragile structure. In comparison, the Butler matrices achieve a low profile and wide scanning angle, making them a better option than lens antennas. However, Butler matrices face challenges when it comes to exciting a large number of antenna array elements [33,37,38].

In this paper, wide-angle beam steering is presented for the flat-panel antenna by a closed-form pillbox platform. To simplify the mechanical set-up and integrate appropriately with RF components for mechanical beam steering purposes, the antenna feed is fixed, while beam steering is achieved by rotating a metallic disc in front of the fixed feed. With a fixed feed, it avoids placing integrated horns in the focal plane of the parabolic reflector. A novel quasi-optical system is introduced to smoothly transition quasi-TEM mode propagation to the planar wavefront. First, to confine electromagnetic waves, an SIW-based horn with an extended aperture is attached to the parabolic reflector to create a closed-form platform. Second, a single coupling slot is created alongside and without spacing from the parabolic reflector to ensure an effective coupling and extend the bandwidth.

The remainder of this paper is organized as follows. Section 2 presents the overall design mechanism and discusses the individual part concept, while Section 3 provides and discusses the antenna performance in simulation and experimental environments and comparison with state-of-the-art designs. Finally, the paper is summarized in Section 4.

## 2. Flat Panel Antenna Design

The configuration of the flat panel antenna terminal is presented in Figure 2a, while the detailed structure is shown in Figure 2b,c. The antenna is stacked up with three double-sided Rogers RO-4003C substrates with a thickness (t) of 1.52 mm and a dielectric constant ($\epsilon_{r}$) of 3.55 (see Figure 2b,c). To achieve SIW feeding, the bottom substrate (D_1_) is in the shape of a rectangle and has two metallic layers on both sides (M_1_ and M_2_) to configure the parabolic structure of the pillbox. The middle substrate (D_2_), on the other hand, has a ground plane for metallic patterns (M_2_), as well as a coupling slot on the metallic structure (M_3_) on the top of the substrate. The top substrate (D3) is shaped like a disc and contains metallic strip patterns that work as an antenna array (M4). In principle, the antenna has two main parts: (a) the radiating part (printed on the top layer (D3)), which consists of a rotating disc containing metallic patterns, and (b) the feeding part (a pillbox feeding system), which includes SIW feeding, a coupling slot (etched in the middle layer), and a parabolic reflector (etched through metallic vias between the bottom and middle layers). The disc has a radius of 150 mm, while the overall size of the antenna is $340\times 335\times 2\;mm^{3}$. The working principle of the antenna is discussed in Section 2.1.

### 2.1. Design Principles

The antenna is designed based on a substrate-guided grounded (SGG) concept. This technique requires metallic patterns, feed, guiding substrate, and a ground plane (reflector), as depicted in Figure 3. As can be seen from Figure 3a, the metallic patterns are printed on the top side of the substrate, whereas the ground plane is located on the bottom side of the substrate. The metallic patterns are excited via the waveguide port in the simulator CST microwave studio (see Figure 3b). The waveguide port generates a plane wave, as shown in Figure 3b, which is guided through the substrate and hits the metallic patterns from the edge of the substrate. Once the metallic patterns are excited through the plane wave, it starts radiating the waves into the air, as shown in Figure 3c. This type of radiation is referred to as leaky waves. This structure provides highly directive radiation patterns with low-side lobes, as shown in Figure 3c. Beam scanning is accomplished through the utilization of the mechanical rotation of metallic patterns. Thus, the metallic patterns in the proposed design are printed in a circular shape. The final design can scan the beam from −60° to+60°, which is an extraordinary performance compared to the existing literature [4,35,39,40]. The detailed results are presented in Section 3.

### 2.2. Final Design

In order to develop the antenna based on the SGG method, a circular disc with a radius of 150 mm and a pillbox feeding (quasi-optical) system are developed. Figure 4 shows the details of each layer in the proposed design before assembling the antenna structure. The proposed antenna consists of four conducting layers (M_1_:M_4_) interleaved by three dielectric substrates (D1:D3). Table 1 shows the parameter of the antenna where all the dimensions are in (mm).

#### 2.2.1. Radiating Disc

Linear periodic metallic patterns in a circular manner are printed on a disc to form a top layer, as depicted in Figure 4 (M_4_). The disc is separated from the other structure to produce beam steering by its rotation (D_3_ with M_4_). The configuration of metallic patterns can be chosen based on the required beam direction; it works as an antenna array. Thus, the critical factors are the number of strips, distance between them, and width of each strip. The space between the elements is chosen to avoid any grating lobes and achieve high gain. By changing the shape and size of the metallic patterns, the beam can be varied. To demonstrate this idea, metallic patterns are printed on the top side of the disc. After careful optimization of the patterns, a period of 8 mm and a total length of 15 λ along the x-direction at 20 GHz are chosen. A width of 4 mm and a spacing of 4 mm, which provide beam pointing at the phi=0° and theta=8° direction, are considered.

#### 2.2.2. Feeding Structure

To make the antenna compact, pillbox-based feeding is implemented underneath the disc, as shown in Figure 2. A detailed view of the feeding mechanism along with the used substrates is described in Figure 5. The feeding structure consists of an SIW horn feeding, parabolic reflector, and coupling slot, as shown in Figure 4. The geometric properties of the parabolic reflectors offer the transformation of a cylindrical wave originating from the paraboloid’s focus into a plane wave directed along the parabola axis. Therefore, the feeding structure and the coupling slot are arranged and placed in different layers to avoid aperture blocking. The feeding is positioned in the focal plane of the parabolic reflector located on the lower substrate (D_1_), which is integrated with the horn and SIW structure to feed and guide the waver, respectively. Meanwhile, the coupling slot is placed on the upper substrate (D_2_). Connectivity between the two layers is facilitated through this coupling slot. The reflector surfaces are actualized using vertical metallic vias, establishing a connection between the top metal layers M_1_ and M_3_. Additionally, a coupling slot with a width of 5 mm is etched onto the middle metal layer (M_2_). The SIW horn, which has dimensions of $L_{1}\;and\;L_{2}$, is fed by the 2.92 mm millimeter-wave 50 Ω connector and generates a quasi-cylindrical wave (see Figure 6a). The wave propagates via substrate 3 (Figure 5) towards the parabolic reflector. The parabolic reflector is also SIW-based and converts the incoming wave into a plane wave. The parabolic reflector is generally designed using [34,41]:(1)$$r=\frac{2F}{1+cos\phi }$$
where r is the parabolic surface radius, F refers to the focal length, and ϕ indicates the angle between r and  F vectors (see Figure 5b). The parabolic slot is positioned close to the parabolic surface for maximum power coupling. The coupling slot transforms that plane wave to the top layer (substrate 1), which has metallic patterns; see Figure 6b. The coupling slot, which has a width of $s_{w}$, is placed just on the edge of the parabolic reflector. $s_{w}$ is a critical parameter to improve the impedance-matching bandwidth. The optimized value of the $s_{w}$ is 5 mm for the final design. The SIW horn and parabolic reflector are made of metallic vias that have a radius of 0.5 mm. The focal length and curve aperture are optimized to achieve a low side lobe level. The parabolic reflector has a curve aperture of 301 mm and focal point distance (F) of 126 mm.

Figure 6 shows the electric field distributions of the antenna for the top layer (radiating strips) and bottom layer (pillbox feeding) to help understand its working mechanism. Pillbox feeding generates a plane wave in the x-direction, which excites the radiating strips as a leaky-wave antenna. This excitation produces a broadside radiation pattern when there is no rotation in patterns. However, when the metallic patterns are rotated by an angle ω=±20°, the beam can be steered in azimuth and elevation planes. This can be observed from the electric field distributions in Figure 6c,d. Specifically, to achieve beam scanning, the intersection of the plane wave and the radiating metallic strips is changed by rotating the disc, as shown in Figure 6c,d. The disc is rotated (ω) clockwise and anti-clockwise to produce symmetrical beam scanning.

## 3. Antenna Performance and Discussions

A prototype was fabricated to validate the proposed idea. Figure 7 shows the photograph of the top and bottom views of the fabricated layers before the assembly of the proposed antenna. The three layers are fabricated as layer-1, which is the feeding parabolic, layer-2, which is the coupling layer, and layer-3, which is the radiating layer. It should be noted that the design consists of multi-layer substrates. Small holes are made in the corner of the substrates to assemble the whole structure via nylon screws. A 2.92 mm millimeter-wave 50 Ω connector is soldered from the bottom side of the antenna. Layer-1 and layer-2 were assembled together using nylon screws, and then the rotating disc was placed on the top side and rotated about its central axis to realize beam scanning, as shown in Figure 8a,b. After assembling the antenna, a vector network analyzer was used to test its reflection coefficients. The antenna’s tested reflection coefficients are shown in Figure 9 in a blue-dotted–dashed line. The tested results are also compared with simulations, shown in Figure 9 as a red solid line. It can be seen from the measured results that the 10 dB return loss (−10 dB reflection coefficients) bandwidth is from 19.2 GHz to 20.4 GHz. The measured bandwidth is about 1.2 GHz, which is slightly smaller than the simulated one. The discrepancies between the measured and simulated reflection coefficients are mainly caused by fabrication, soldering, and measurement errors. It should be noted that the antenna has multiple layers and is assembled in the laboratory; this type of discrepancy is expected due to the manual assembly of the layers in addition to the human factor error of the soldering, where the soldering point can work as an additional stub at a higher frequency, which induces mismatch between the antenna and the connector. This can be minimized by having more accurate assembly of the layers, reducing the air gap between layers, and minimizing measurement errors.

The mechanical rotation is manually realized by rotation steps of ω=5° from −20° to 20°. The measurement setup for ω=20° is shown in Figure 8c. The antenna is tested for radiation patterns and realized gain values. To understand the beam steering of the proposed antenna, the normalized measured radiation patterns are shown in Figure 10. It is evident that the beam direction is steered with a little gain variation (~3 dBi) over the whole rotating angle. The radiation patterns of nine different rotation angles have been plotted in Figure 10 from −20° to 20° with a step of 5°. The intermediate coverage area between the beams can be covered by selecting small intermediate rotation angles. It is noted that the antenna offers 120° as a scanning angle, while Figure 11 depicts the 2D plots of the measured radiation patterns at 20 GHz for the nine rotation angles for the whole domain 0≤θ≤180°;−180°≤ϕ≤180°. The plots demonstrate the ability of the antenna to achieve beam steering with high gain and low side lobes. On the other hand, one of the essential merits of evaluating the scanning area of the antenna is the total scanning angle calculated from the total scan pattern (TSP) [42,43,44]. The TSP is the maximum achievable gain at each certain angular angle $\left(\theta ,\phi \right)$ in the whole spherical coordinate, where the TSP is computed for all potential antenna rotation angles ($\omega_{1}:\omega_{n}$) and then the maximum gain (G) is selected at each point in the spatial distribution as [44]:(2)$$TSP\left(\theta ,\;\phi \right)=\max\left[G_{\omega_{1}}\left(\theta ,\;\phi \right),\;{G_{\omega_{2}}\left(\theta ,\;\phi \right),\;\ldots \ldots \ldots \;,\;G}_{\omega_{n}}\left(\theta ,\;\phi \right)\right],\;0\leq \theta \leq 180{}^{\circ};-180{}^{\circ}\leq \phi \leq 180{}^{\circ}$$

As shown in Figure 11, the measured $TSP\left(\theta ,\;\phi \right)$ demonstrates the ability of the antenna to achieve beam steering of 120° with a tiny variation in the beam in the elevation plane (±4°) and without any null, which assembles only from nine rotations of the disc within ±20°. The simulated and measured gain values of the steering beam at the different rotating angles are shown in Table 2. The maximum measured gain is 23.2 dBi at w=0° rotation, which is 3.3 dB less than the simulated one. This difference in gain values is expected because of higher losses of the substrate at a high frequency. It is also attributed to the fabrication, soldering, air gaps, and measurement errors. Nevertheless, this is acceptable for large antennas, especially those operating at high-frequency bands. The variations in the measured gain from −40° to+40° scanning range is only 2 dBi, which is better than the other reported designs [21,36], while a 3.3 dBi  variation is observed in the whole scanning range from −60° to +60°.

In order to determine the quality of the antenna and its performance, it is compared to other reference antennas in terms of antenna height, frequency band, beam coverage, measured gain, and SLLs at the broadside and maximum scan angle. These characteristics are given in Table 3. By rotating a horn within the focal plane of the pillbox feeding system, the reference antenna that was described in [39] possesses a high gain and low SLL when viewed from the broadside. However, it displays a high SLL when viewed from a maximum scan angle of ±40 degrees. Additionally, the antenna height is $4.39\lambda_{0}$ which is much higher than the proposed design. Furthermore, it features a mechanically steered main beam that extends beyond ±40 degrees. The other reference antenna that is presented in [21] has a low SLL at the broadside and a maximum scan angle of 40 degrees; however, the measured gain is low (i.e., 19.6 dBi), and the beam coverage that is achieved is only ±40 degrees. Although the antennas in [35,36,40] offer a small profile, they suffer from either a limited coverage angle or high side lobe level. The antennas in [39,45] have a wider bandwidth compared to the others due to using high profiles ($4.39\;\lambda_{0}$ and $9\;\lambda_{0}$). However, they offer limited angle scanning and high SLL. The antenna proposed in this study, on the other hand, possesses a relatively high gain, which is, i.e., 23.2 dBi. Additionally, it has a low SLL of −17.8 dB at a maximum scan angle of 60 degrees. Furthermore, the antenna height is only $0.3\;\lambda_{0}$, and it has a beam coverage of ±60 degrees on the azimuth plane.

## 4. Conclusions

In this work, wide-angle beam steering has been shown for the purpose of on-the-move satellite communication through the utilization of an innovative closed-form pillbox antenna system. The complete wide-angle beam steering antenna system includes three main parts: fixed-feed SIW-based horn, novel quasi-optical system, and leaky-wave radiating part. A transition made by a single coupling slot alongside and without spacing from the parabolic reflector has been proposed for the quasi-optical system to have effective coupling and extend the bandwidth. Through the utilization of the mechanical rotation of the metallic patterns, beam scanning is accomplished. The experimental results show that the antenna can provide a beam-scanning capability of 120° in the azimuth plane. The antenna terminal demonstrates directive radiation pasterns with a maximum measured gain of 23.2 dBi with an antenna height of $\;0.3\lambda_{0}$ and low SLL of −17.8 dB at a maximum scan angle of 60°. The antenna operates in the frequency range from 19.2 GHz to 20.4 GHz with a reflection coefficient less than −10 dB. This concept has the potential to be extended to achieve circular polarization, which might be accomplished by incorporating a superstrate layer of a thin polarizer.

## Figures and Tables

**Figure 1 sensors-24-00732-f001:**
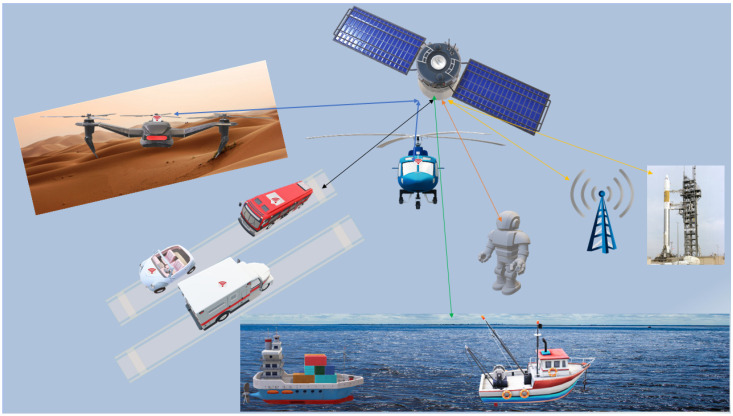
The conceptual scenario of satellite communications on-the-move services.

**Figure 2 sensors-24-00732-f002:**
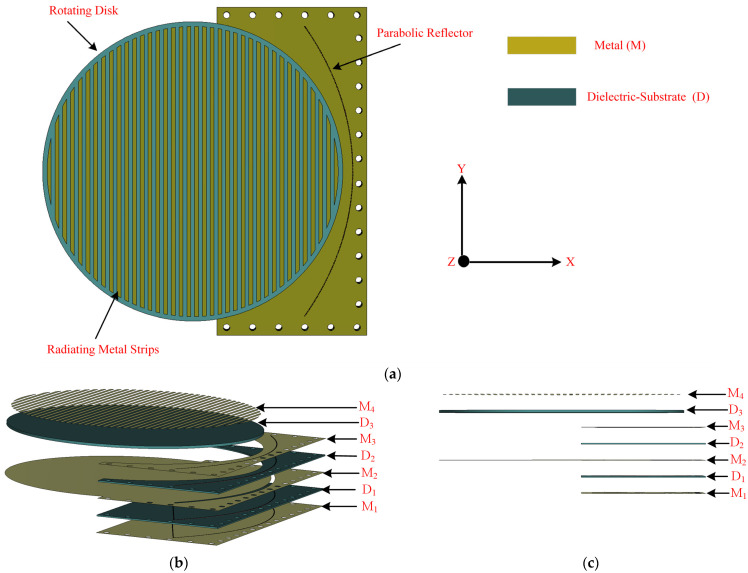
(**a**) The geometry of the proposed antenna and (**b**) its detailed configuration showing multiple layers and (**c**) side view of the layers, where M and D refer to metal and dielectric layers, respectively.

**Figure 3 sensors-24-00732-f003:**
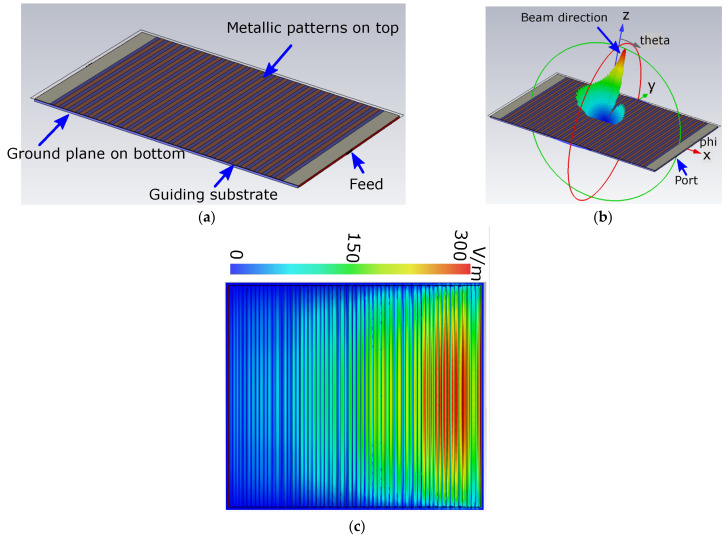
Antenna utilizing substrate-guided grounded concept. (**a**) Configuration of the antenna metallic patterns excited from waveguide port, (**b**) electric field distributions, and (**c**) simulated 3D highly directive radiation patterns obtained at 20 GHz.

**Figure 4 sensors-24-00732-f004:**
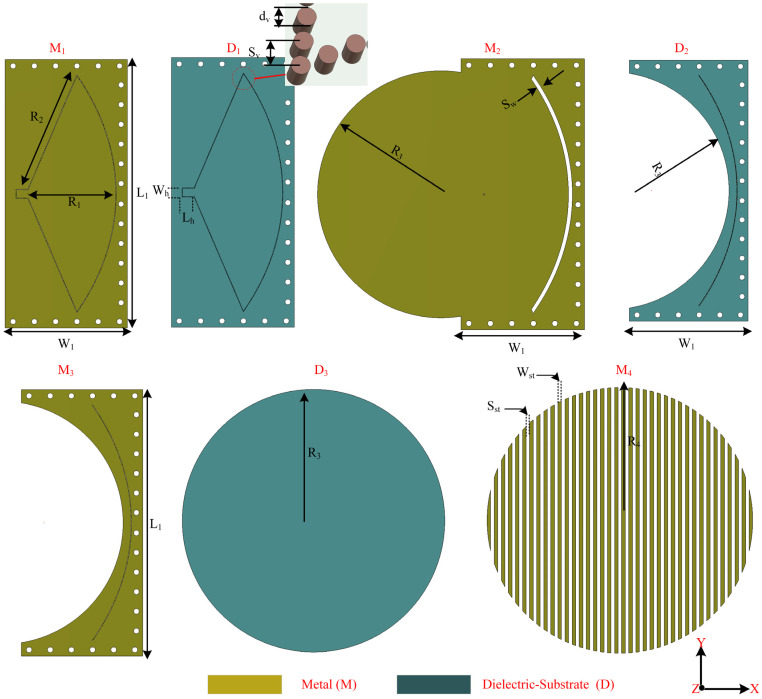
The layers’ details of the proposed design, M and D, refer to the metal and dielectric layer, respectively. There are three dielectric layers and four metallic layers.

**Figure 5 sensors-24-00732-f005:**
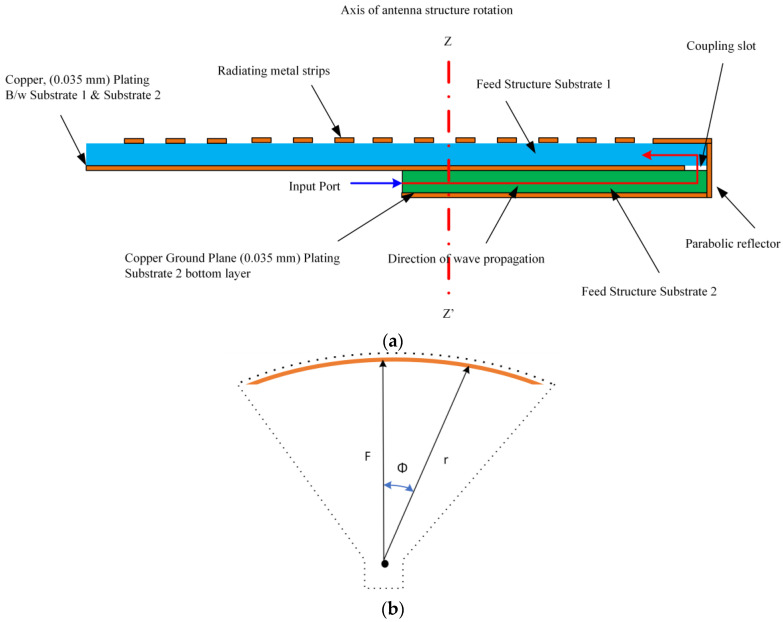
(**a**) Detailed view of layers and material stack up for the antenna and (**b**) pillbox schematic.

**Figure 6 sensors-24-00732-f006:**
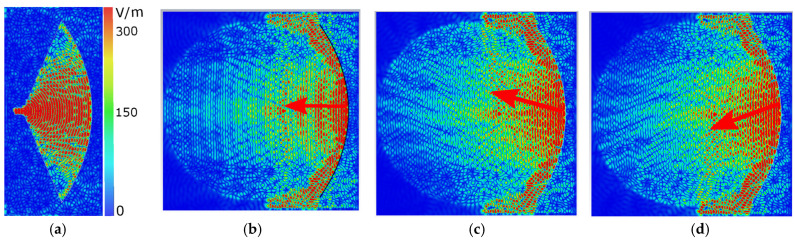
The electric field distributions obtained at 20 GHz. (**a**) The bottom layer and the top layer (**b**) without any rotation *ω* = 0°, (**c**) at the maximum scanning angle on the right side *ω* = 20°, and (**d**) at the maximum scanning angle at the left side *ω* = −20°. Red arrows are pointing in the direction of the propagation of waves.

**Figure 7 sensors-24-00732-f007:**
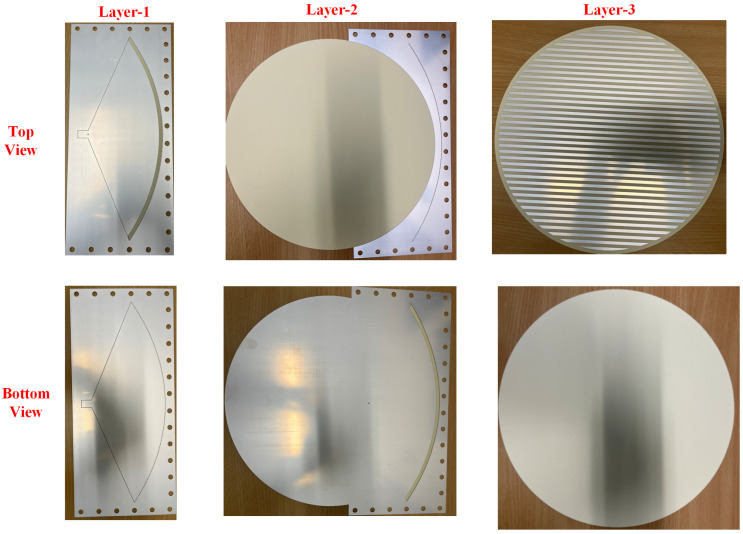
Photographs of the top and bottom views of the fabricated layers before the assembly. Layer-1 is the parabolic feed structure, layer-2 is the coupling layer, and layer-3 is the radiating strips layer.

**Figure 8 sensors-24-00732-f008:**
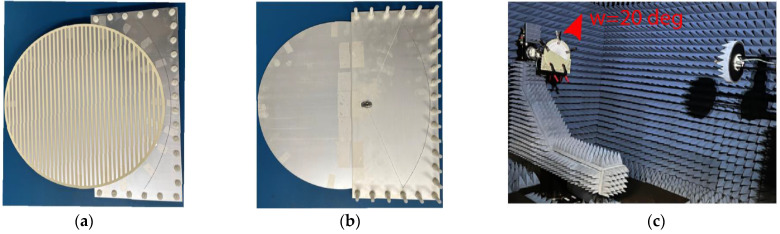
Photographs of the fabricated prototype. (**a**) Top view, (**b**) bottom views, and (**c**) measurement setup with a disc rotation of 20°.

**Figure 9 sensors-24-00732-f009:**
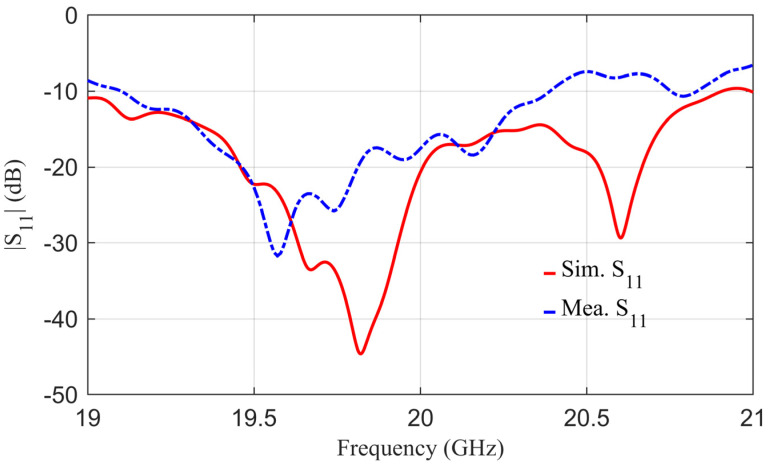
The simulated and measured reflection coefficient of the proposed antenna versus frequency.

**Figure 10 sensors-24-00732-f010:**
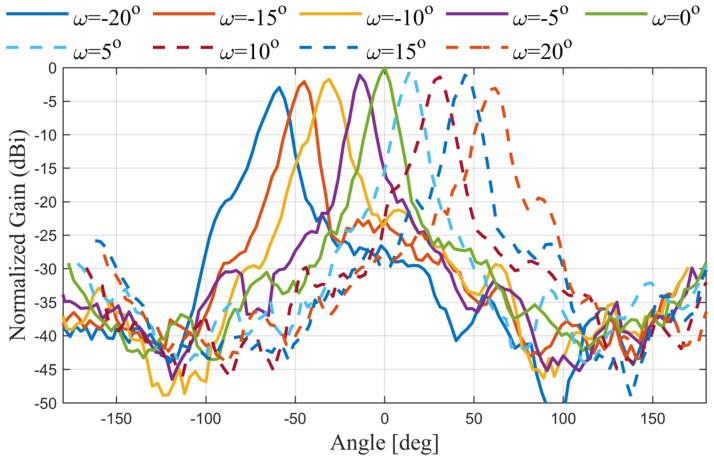
The measured radiation patterns of the antenna with the rotation angles from −20° to 20°. All the radiation patterns are normalized to the maximum gain at *ω* = 0°.

**Figure 11 sensors-24-00732-f011:**
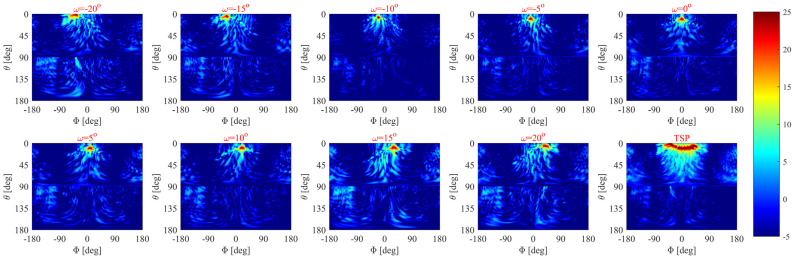
Two-dimensional plot of measured radiation patterns in dBi at rotation angles from −20° to 20° (with 5° spatial angle) and the total scan pattern of those rotation angles. All the plotted values present the gain at a whole spherical angle for each rotated angle, and then TSP is calculated based on (2).

**Table 1 sensors-24-00732-t001:** Dimensions of the proposed antenna; all values in mm.

L_1_	W_1_	R_1_	R_2_	R_3_	R_4_	L_h_	W_h_	S_v_	d_v_	S_w_	W_st_	S_st_
340	115	111.4	157.2	155	150	16	12	1.5	1	5	4	4

**Table 2 sensors-24-00732-t002:** The simulated and measured peak gain at 20 GHz for different rotation angles.

Angle of Rotation *ω*	−20°	−10°	0°	10°	20°
Simulated Gain (dBi)	24	25.5	26.5	25.5	24
Measured Gain (dBi)	20.21	22.1	23.2	22.5	20.18

**Table 3 sensors-24-00732-t003:** Performance comparison of the proposed and previous quasi-optical antenna system.

Ref.	Antenna Height *λ*_0_	BW (GHz)	Beam Coverage (Degree)	Measured Gain (dBi)	SLL (dB)	
					@ BS	@ MSA
[35]	0.08	23.9–24.9	±36	17.7	−15	−8.8
[40]	0.08	23.5–25.7	±30	23.8	−12	−16
[39]	4.39	27.5–31.0	±40	28.9	−32	−11
[36]	0.08	23.5–24.6	±40	25	−15	−10
[21]	0.32	10.2–13.9	±40	19.6	−30	−15.6
[45]	9	28.5–31.0	±24	31.3	−20	−10
[46]	0.1	23.8–24.3	±39	21.6	−23	−10
[47]	0.1	23.8–24.4	±40	24.2	−24	−11
[48]	0.1	23.5–24.5	±35	22	−20	−15
[49]	0.25	74–78	±40	24	−25	−10
This Work	0.3	19.0–20.4	±60	23.2	−20	−17.8

SLL: side lobe level, BW: bandwidth, BS broadside angle, MSA: maximum scanning angle.

## Data Availability

All data has been included in the paper.

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
