# Peer review of "Wide-Angle Beam Steering Closed-Form Pillbox Antenna Fed by Substrate-Integrated Waveguide Horn for On-the-Move Satellite Communications"

_sensors, 2024, doi:10.3390/s24030732_

Round 1

Reviewer 1 Report

Comments and Suggestions for Authors

The paper focuses on the antenna beam control technology of low orbit satellite communication systems and has clear engineering application value. On the basis of previous research, the author proposes a wide-angle mechanical beam steering system for mobile satellite communication based on a closed bunker antenna system. It includes three main parts: a fixed fed SIW horn, a new quasi optical system, and a leaky wave radiation part. The author provided a detailed introduction to the design scheme and simulation results of the system, and measured the processed object. The actual measurement and simulation results both indicate that the scheme is practical and feasible.

Compared to the design schemes proposed by peers, the advantage of this scheme is that the antenna designed by the author has a height of only 0.3 times the wavelength, relatively high gain, and low standing wave. The beam coverage in the azimuth plane is ± 60 degrees.

Shortcomings: The author provided a detailed introduction to the design scheme, but lacked theoretical analysis. It is recommended that the author supplement theoretical analysis.

Author Response

1. Summary

Comments 1:

Shortcomings: The author provided a detailed introduction to the design scheme, but lacked theoretical analysis. It is recommended that the author supplement theoretical analysis.

Response 1: Thank you for your positive feedback about the paper. The theoretical analysis of the Pillbox design has been added to section 2.2 in the revised paper.

Reviewer 2 Report

Comments and Suggestions for Authors

A novel closed-form pillbox antenna system has been presented for on-the-move satellite communication. The beam scanning is achieved using mechanical rotation of the metallic patterns.

The experimental results show that the antenna can provide a beam scanning capability of 120𝑜 in the azimuth plane and 12𝑜 in the elevation plane. The antenna terminal demonstrates directive radiation pasterns having a maximum measured gain of 23.2 𝑑𝐵𝑖 with an antenna height of 0.3𝜆𝑜 and low SLL of −17.8 𝑑𝐵 at a maximum scan angle of 60𝑜.

Thanks

Author Response

Thank you very much for taking the time to review this manuscript and for accepting the paper.

Reviewer 3 Report

Comments and Suggestions for Authors

1.In the article a wide-angle beam steering has been presented for on-the-move satellite communication. A novel closed-form pillbox antenna system is examined.

2.The above-mentioned system consists of three parts: (i) fixed- feed SIW-based horn, (ii) novel quasi-optical system, and (iii) leaky-wave radiating part.

3.In the paper, a wide-angle beam steering is presented for the flat-panel antenna by a closed-form pillbox platform.

4.The suggested antenna terminal is suitable for next-generation ubiquitous connectivity for households and small businesses in remote areas, ships, unmanned aerial vehicles, and disaster management.

5.The experimental results show that the antenna can provide a very good productivity.

6.The antenna terminal demonstrates high effectiveness.

7.In general the article is prepared at a good level (all parts including introduction material, description of the illustrations, conclusion, the list of references). The illustration material will be very useful for readers.

8.It will be reasonable to add a brief description of the prospective future research directions.

Author Response

1. Summary

Thank you very much for taking the time to review this manuscript and thank you for your positive feedback and for accepting the paper.

Comments 1: It will be reasonable to add a brief description of the prospective future research directions.

Response 1: The prospective future work directions have been added to the conclusion of the revised paper.